# The rostral intralaminar nuclear complex of the thalamus supports striatally mediated action reinforcement

Kara K Cover, Abby G Lieberman, Morgan M Heckman, Brian N Mathur*

Department of Pharmacology, University of Maryland School of Medicine, Baltimore, United States

**Abstract** The dorsal striatum (DS) mediates the selection of actions for reward acquisition necessary for survival. Striatal pathology contributes to several neuropsychiatric conditions, including aberrant selection of actions for specific rewards in addiction. A major source of glutamate driving striatal activity is the rostral intralaminar nuclei (rILN) of the thalamus. Yet, the information that is relayed to the striatum to support action selection is unknown. Here, we discovered that rILN neurons projecting to the DS are innervated by a range of cortical and subcortical afferents and that rILN→DS neurons stably signaled at two time points in mice performing an action sequence task reinforced by sucrose reward: action initiation and reward acquisition. In vivo activation of this pathway increased the number of successful trials, whereas inhibition decreased the number of successful trials. These findings illuminate a role for the rostral intralaminar nuclear complex in reinforcing actions.

## Editor's evaluation

Cover et al. examine the pathway from the intralaminar nucleus of the thalamus (rILN) to the dorsal striatum (DS) in the reinforcement of behavior/actions. The rILN sends a large glutamatergic projection to the DS, but its role in action selection was unknown. The authors found that the rILN neurons that project to the DS were activated at both action initiation and with the reward, and that activation and inhibition of this pathway increased the success or decreased the success of reward acquisition, respectively. The findings are an important advance in our understanding of the function of rILN to DS projection in reward-based behavior, and the manuscript has provided convincing evidence with the appropriate methodologies to support these claims.

*For correspondence:
bmathur@som.umaryland.edu

## Introduction

Classic models of basal ganglia function regard the dorsal striatum (DS) as an integrator of cortical glutamatergic signaling that is modulated by reward signals arising from midbrain dopaminergic input to support the reinforcement of actions directed toward the acquisition of rewards necessary for survival (*Albin et al., 1989*; *DeLong and Wichmann, 2009*; *Gerfen and Surmeier, 2011*; *Kreitzer and Malenka, 2008*). While several studies highlight the fact that thalamic nuclei directly project to the striatum (*Alloway et al., 2014*; *Ding et al., 2008*; *Parker et al., 2016*; *Smith et al., 2014*), these nuclei are far less represented in basal ganglia models. This reflects our relatively poor understanding of the functional role of thalamostriatal signaling in action reinforcement.

The intralaminar nuclei are the primary source of thalamic excitatory input to the striatum (*Elena Erro et al., 2002*; *Parent et al., 1983*). These nuclei are segregated into rostral and caudal subdivisions. At the caudal end, the medullary lamina splits to create the parafascicular nucleus, and more

laterally, the centér median nucleus in monkey and human brains. The boundary distinguishing these two nuclei is undetectable in rodents and other smaller mammals; thus, the posterior intralaminar nuclei are referred to solely as the parafascicular nucleus with the consideration that the lateral component of this nucleus is homologous to the centér median nucleus (*Jones, 2007*). At the rostral end, the so-called rostral intralaminar nuclei (rILN) include the poorly described and weakly demarcated central lateral (CL), paracentral (PC), and central medial (CM) regions. Both the caudal intralaminar grouping and the rILN receive input from a wide array of cortical and subcortical centers and output to both the striatum and the cortex. As such, it is unclear which of the inputs to the intralaminar nuclei are routed to the striatum to potentially influence action reinforcement (*Groenewegen and Berendse, 1994*).

Functionally, the caudal intralaminar nuclear group responds to salient sensory cues in monkey (*Minamimoto and Kimura, 2002*) and, in rodents, detects changes in action-outcome contingencies to facilitate behavioral flexibility (*Bradfield et al., 2013*; *Bradfield and Balleine, 2017*; *Yamanaka et al., 2018*). Recent work implicates the rILN in facilitating the execution and switching of learned actions (*Kato et al., 2018*) and driving behavioral reinforcement (*Cover et al., 2019*; *Johnson et al., 2020*). Together, these data suggest that the rILN contribute to action reinforcement, but the information that the rILN integrates and relays to the striatum to support this function is unknown. Examining this in mice using in vivo neural circuit monitoring and manipulation methods, we discovered that the rILN dynamically signal at both the initiation of an action and during reward acquisition to optimize action performance. We also determined that striatal-projecting rILN neurons demonstrate homogeneous physiological properties across the three regions but differently integrate cortical, midbrain, and hindbrain information that is then passed on to the striatum. These data support the notion that the rILN are a central integrator contributing to basal ganglia-mediated action reinforcement.

## Results

### Striatal-projecting rILN neurons exhibit uniform properties

The rILN consist of a contiguous band of cells within the internal medullary lamina that is parcellated into CL, PC, and CM nuclei (*Berman et al., 1983*). Although anatomical tracing suggests subtle differences in afferent and efferent connectivity among these three subregions (*Van der Werf et al., 2002*), it is unclear whether these nuclei represent functionally distinct cellular groups. To investigate whether rILN→DS neurons exhibited physiological differences across the three nuclear divisions, we injected a retrograde traveling tdTomato-expressing virus in the central DS of mice and assessed neurophysiological properties of rILN→DS neurons using whole-cell patch clamp electrophysiology in acute slices (*Figure 1A and B*). We found that neurons across the three nuclei did not significantly differ in membrane capacitance (*Figure 1C*; F[2,40]=1.242, p=0.30), membrane resistance (*Figure 1D*; F[2,40]=1.212, p=0.31), input resistance (*Figure 1E*; F[2,40]=0.520, p=0.60), or resting membrane potential (*Figure 1F*; F[2,40]=0.124, p=0.88).

To assess differences in intrinsic firing properties, we injected a current ramp to induce action potential (AP) firing. We did not observe a significant difference in the AP threshold between the three nuclei (*Figure 1G*; F[2,40]=2.684, p=0.08). Following a series of 0.5 s current steps, the maximum firing rate, as calculated from the first six APs, did not differ between neurons from the three rILN nuclei (*Figure 1H*; F[2,39]=1.411, p=0.26). AP accommodation was also not different between the nuclei (*Figure 1—figure supplement 1A*; F[2,35]=1.671, p=0.20). We observed that hyperpolarizing current steps induced post-hyperpolarization burst firing in all three nuclei (as shown in *Figure 1C*). We did not find differences in either the post-hyperpolarization burst firing inter-spike interval (*Figure 1— figure supplement 1B*; F[2,35]=1.003, p=0.38) or AP number (*Figure 1—figure supplement 1C*; F[2,40]=2.837, p=0.07).

### rILN→DS neurons signal at action initiation and reward acquisition

Establishing that rILN→DS neurons display homogeneous intrinsic electrophysiological properties, we next endeavored to elucidate when this pathway is active during action learning and performance. We fiber photometrically recorded activity-dependent calcium signaling selectively from rILN→DS neurons in mice learning a lever-pressing operant task (*Figure 2A*). To assess whether rILN→DS neurons respond to sensory stimuli eliciting an action, action initiation, termination, kinematic properties, or reward, we trained mice to respond to the lever extension by pressing five times in a defined

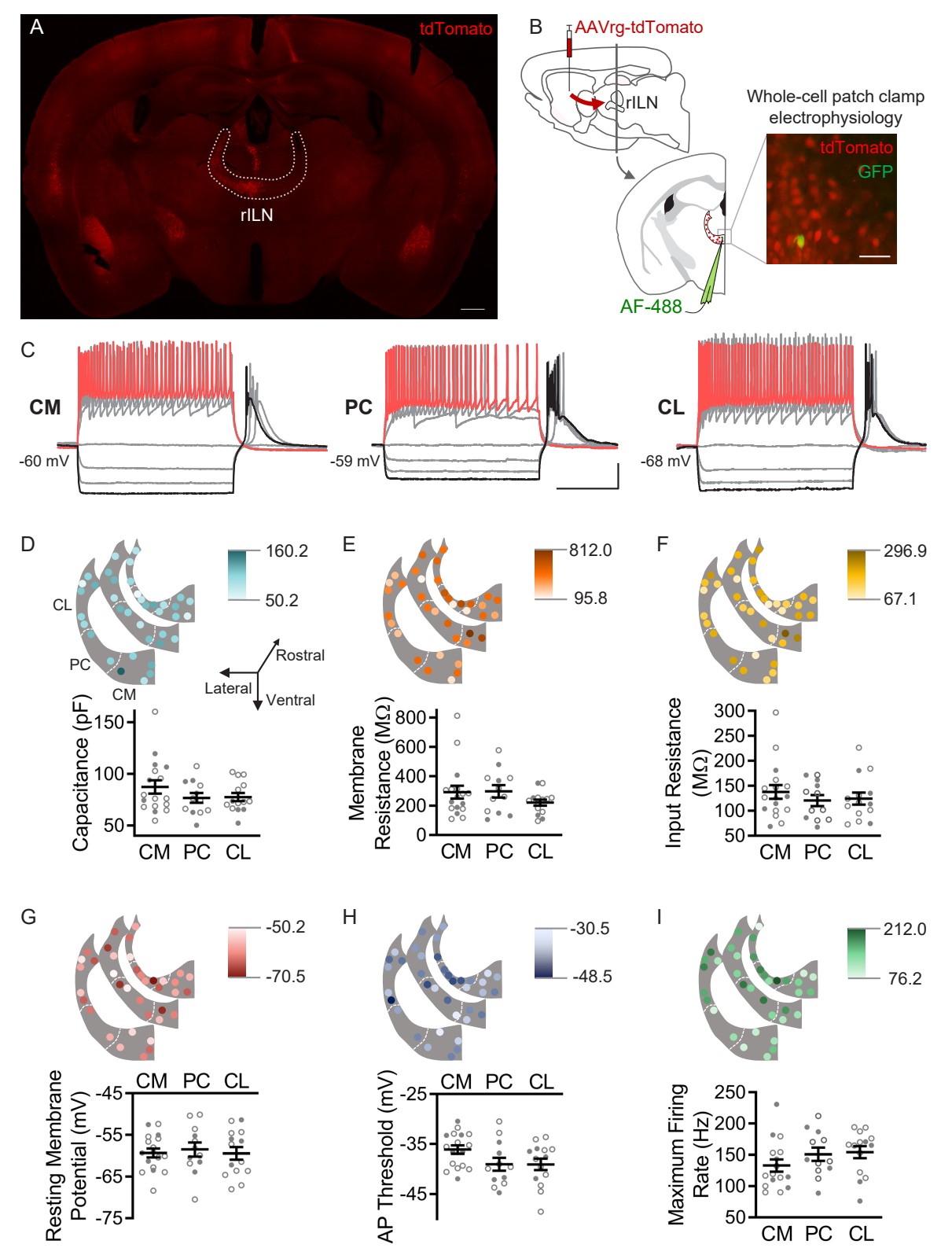

**Figure 1.** Dorsal striatum (DS)-projecting neurons of the rostral intralaminar nuclei (rILN→DS) thalamus exhibit uniform physiological properties. (**A**) rILN→DS neurons project ipsilaterally. Coronal mouse brain section through the rILN. Following AAVrg-tdTomato injection in the left DS left rILN→DS neurons labeled with tdTomato. (**B**) Left: schematic of experimental strategy to label rILN→DS neurons. Right: a patched neuron (GFP, injected with AlexaFluor[AF]–488) among tdTomato-labeled rILN neurons. (**C**) Representative traces showing central medial (CM), paracentral (PC), and central lateral

*Figure 1 continued on next page*

*Figure 1 continued*

(CL) neuronal responses to 500 ms current injection steps. (**D**) Top: neuronal membrane capacitance of rILN neurons mapped across three rostral-caudal coronal planes in light to dark color gradient spanning minimum to maximum capacitance values. Bottom: membrane capacitance did not differ between CM (n=17), PC (n=12), and CL (n=13) nuclei. (**E–I**) Analysis of membrane resistance (**E**), input resistance (**F**), resting membrane potential (**G**), action potential (AP) threshold (**H**), and maximum firing rate elicited by current steps (**I**; CM n=16), for which no significant differences existed between the three rILN nuclei. Scale bars: 500 μm (**A**), 100 μm (**B**), 20 mV and 200 ms (**C**). One-way ANOVA (**D–I**). n=number of cells. Filled and open data points represent cells from male and female mice, respectively. Error bars = standard error of the mean (SEM).

The online version of this article includes the following source data and figure supplement(s) for figure 1:

**Source data 1.** Source electrophysiological data for *Figure 1*.

**Figure supplement 1.** Additional rostral intralaminar nuclei (rILN) firing properties.

**Figure supplement 1—source data 1.** Source electrophysiological data for *Figure 1—figure supplement 1*.

period of time for a sucrose pellet reward (*Figure 2B*). In early training stages, mice must complete one, three, or five fixed-ratio presses (FR1, FR3, and FR5, respectively) with no time limit (NTL) for completion. Intermediate schedules consist of progressively shortened response periods to complete the FR5, terminating at a 5 s time limit. We found transient increases in rILN→DS activity that accompanied the first press in the action sequence across all stages of training (*Figure 2C*). Comparing the average rILN→DS activity z-score at 1.5 s before and at the time of the first press, we observed significant differences from the first training protocol (fixed ratio 1; FR1), an intermediate schedule (FR5-NTL), and the final training schedule (FR5-5s time limit; time relative to press: $F_{[1,12]}=26.73$, p=0.0002; schedule: $F_{[2,24]}=9.587$, p=0.0009). We observed similar positive fluctuations in rILN→DS activity aligned to subsequent presses in early training protocols in which mice complete the FR lever-pressing task over an extended period (*Figure 2D–F*). This activity was not observed for subsequent presses in mice on the terminal training protocol, however (*Figure 2D–F*; schedule: $F_{[1,12]}=34.79$, p<0.0001; press: $F_{[3,36]}=2.146$, p=0.11).

We next examined how rILN→DS neurons activate relative to specific elements of the FR5 task. We found that rILN→DS activity peaked following the extension of the lever into the operant chamber on completed trials in intermediate and terminal stages of training and that the magnitude of this signal change increased with training (*Figure 2G*; time relative to lever: $F_{[1,12]}=17.27$, p=0.0013; schedule: $F_{[3,36]}=14.49$, p<0.0001; interaction: $F_{[3,36]}=6.776$, p=0.001). The average latency to initiate the lever press for completed trials was 8.7 s for FR1 and decreased to 1.9 s (FR5-NTL) and 0.76 s (FR5-5s; *Figure 2H*; $F_{[1.218,14.61]}=303.0$, p<0.0001). Gross rILN→DS activity at this time, as measured by area under the photometric signal, was negatively correlated with initiation latency on completed trials across all FR5 schedules ($r=-0.161$; –0.185 and –0.137 95% CI, p<0.0001, n=6338 trials). Thus, we examined whether this lever press-associated rILN→DS signal varied by task performance. We found that a larger signal accompanied completed trials as compared to incomplete trials (in which mice pressed one to four times; *Figure 2I*; time relative to first press: $F_{[1,12]}=14.73$, p=0.0024; trial type: $F_{[1,12]}=2.374$, p=0.15).

To further explore this rILN→DS signal, we investigated whether rILN→DS activity correlates with general movement by recording mice freely moving in an open arena (*Figure 2—figure supplement 1A*). We observed increases in rILN→DS signaling aligned to movement onset (*Figure 2—figure supplement 1B*; t=2.817, p=0.037) and preceding the maximum velocity achieved during movement epochs (*Figure 2—figure supplement 1C*; t=4.116, p=0.009). Significant changes in rILN→DS activity did not accompany movement cessation (*Figure 2—figure supplement 1D*; t=0.341, p=0.75).

To discriminate between action initiation and the sensory cues (e.g. lever extension) that may signal action initiation, we used a Pavlovian appetitive conditioning paradigm in which sucrose pellets were administered at the termination of an auditory tone (*Figure 3—figure supplement 1A–B*). We did neither find tone-related changes in rILN→DS signaling nor did we observe learning-dependent changes in either the tone-paired group (*Figure 3—figure supplement 1C*; time relative to tone: $F_{[1,5]}=2.046$, p=0.21; training stage: $F_{[1,5]}=2.466$, p=0.18) or the tone non-paired control cohort (*Figure 3—figure supplement 1D*; time relative to tone: $F_{[1,5]}=2.312$, p=0.19; training stage: $F_{[1,5]}=0.204$, p=0.67).

Alternatively, our observation of training-dependent increases in action initiation-associated rILN activity may reflect the learned expectation of a rewarded action. To test this, we recorded rILN→DS activity after switching FR5 reinforcement to 0%. Mice extinguished FR5 lever pressing within several

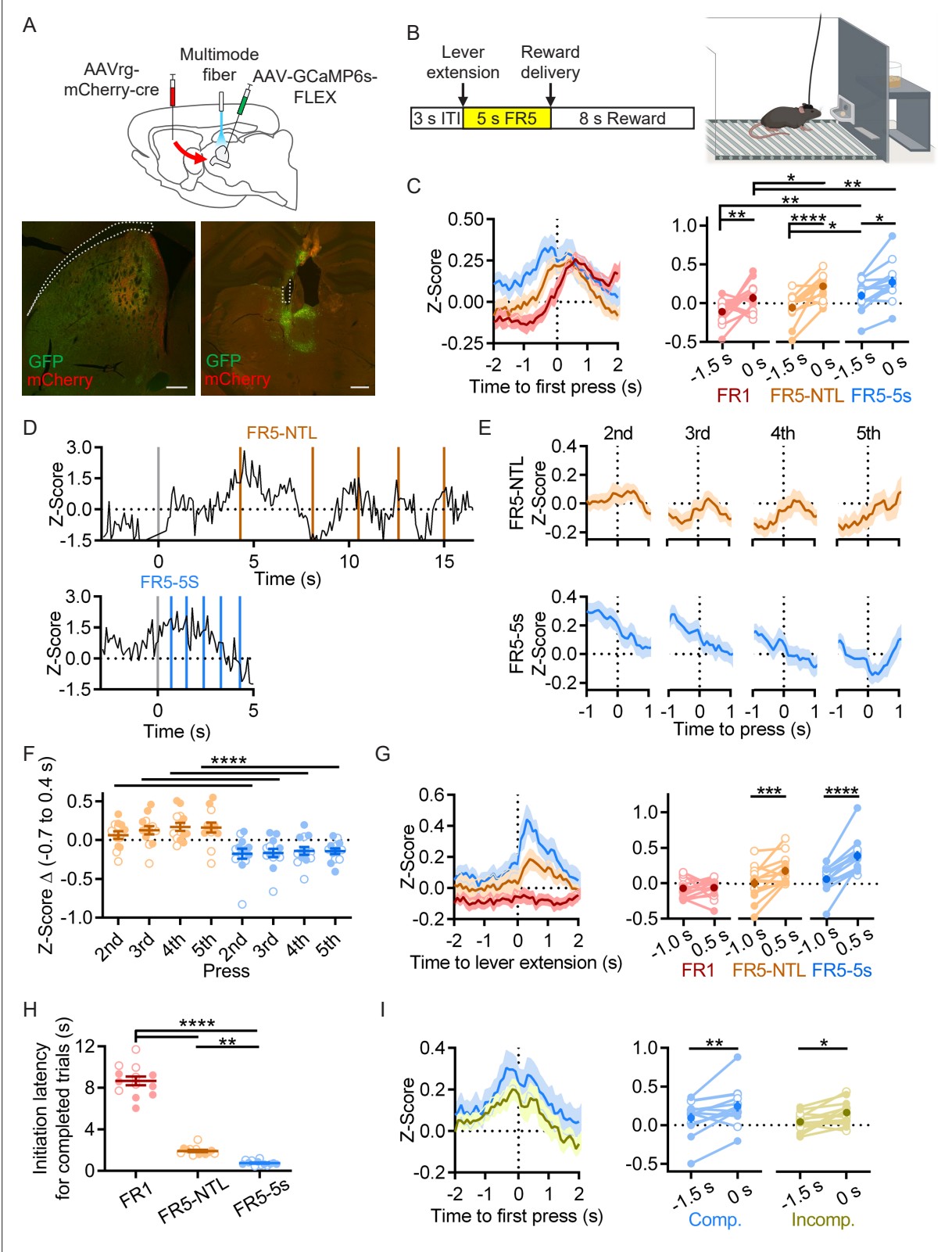

**Figure 2.** Rostral intralaminar nuclei (rILN)→dorsal striatum (DS) activity aligns to action initiation. (**A**) Top: experimental strategy for fiber photometric recording of rILN→DS neurons. Bottom: representative expression of cre recombinase (mCherry) and cre-dependent GCaMP6s (GFP) in the DS (left) and rILN (right) with fiber placement in the rILN (right). (**B**) Left: schematic of lever press trial design. Each trial starts with a 3 s inter-trial interval (ITI) followed by lever extension. A sucrose pellet is delivered following the fifth press, with an 8 s consumption period before the next trial. Right: cartoon of operant

*Figure 2 continued on next page*

*Figure 2 continued*

chamber with lever and sucrose pellet dispenser. (**C**) Left: average calcium-dependent activity of rILN→DS neurons aligned to the first press for the first training schedule (red; Fixed-rate 1 no time limit; FR1-NTL), an intermediate protocol (orange; FR5-NTL), and the terminal training schedule (blue; FR5-5s time limit). Right: rILN→DS activity was significantly greater at the time of the first press compared to 1.5 s prior for all three operant schedules (N=13). (**D**) Representative rILN→DS activity from a mouse completing the FR5 sequence on FR5-NTL (top) and FR5-5s (bottom left) schedules. Gray lines indicate time of lever extension, and colored lines show individual lever presses. (**E**) Average rILN→DS activity for presses 2–5 from FR5-NTL (top) and FR5-5s (bottom) schedules. (**F**) The change in z-score from –0.7 s to 0.4 s (relative to press) was positive for all presses on the FR5-NTL schedule (orange) and negative for all presses on the FR5-5s schedule. (**G**) Left: photometrically recorded rILN→DS activity aligned to the extension of the lever on completed FR5 trials from FR1 (red), FR5-NTL (orange), and FR5-5s (blue) training schedules. Right: rILN→DS signaling increased following lever extension on all schedules except for FR1. (**H**) The average latency to initiate pressing progressively decreased with training. (**I**) Left: first press-aligned rILN→DS activity from completed (blue) and incomplete (yellow) FR5 trials in trained mice. Right: the average rILN→DS activity change from –1.5 to 0 s relative to first press differed by trial performance. Scale bars: 500 μm. Two-way repeated measures ANOVA (**C, F, G, and I**); one-way repeated measures ANOVA (**H**). N=number of mice. Filled and open data points represent male and female mice, respectively. Error bars = SEM. * p<0.05, ** p<0.01, *** p<0.001, and **** p<0.0001.

The online version of this article includes the following source data and figure supplement(s) for figure 2:

**Source data 1.** Source photometric and FR5 data for *Figure 2*.

**Figure supplement 1.** Rostral intralaminar nuclei (rILN)→dorsal striatum (DS) neuronal activity correlates with movement initiation.

**Figure supplement 1—source data 1.** Source movement and photometric data for *Figure 2—figure supplement 1*.

sessions (*Figure 3—figure supplement 1F*, t=11.69, p<0.0001), but the first press-aligned signal on completed (but unreinforced) FR5 trials did not differ significantly from reinforced sessions (*Figure 3—figure supplement 1G*; time relative to lever: F[1,11]=13.36, p=0.0069; reinforcement: F[1,11]=0.1628, p=0.73). In an alternative paradigm, FR5-trained mice alternated between sessions in which completed trials were reinforced at 100 or 50% probability. Lowering the probability of reinforcement did neither alter performance (*Figure 3—figure supplement 1H*; t=0.788, p=0.45) nor did it produce significant differences in first press-aligned rILN activity (*Figure 3—figure supplement 1I*; time relative to press: F[1,10]=5.571, p=0.04; reinforcement rate: F[1,10]=0.304, p=0.59).

Lastly, we observed that rILN→DS activity increases at reward acquisition. Sucrose pellet retrieval was accompanied by increased rILN activity on all FR training protocols (*Figure 3A–B*; time relative to retrieval: F[1,12]=57.24, p<0.0001; schedule: F[1,12]=10.42, p=0.007; interaction: F[1,12]=5.311, p=0.040). Additionally, this signal negatively correlated with reward retrieval latency across all FR5 schedules (*Figure 3C*; r=−0.123, −0.147, and −0.097, 95% CI, p<0.0001). Reward delivery was necessary for this activity, as unreinforced FR5 trials did not elicit changes in rILN→DS activity (*Figure 3D*; time relative to head entry: F[1,12]=21.01, p=0.006; reinforcement: F[1,12]=5.610, p=0.035; interaction: F[1,12]=89.14, p<0.0001). On intermediate training schedules, mice frequently check the pellet receptacle in between individual presses. We examined rILN→DS activity during these 'premature' receptacle head entries on trials that were ultimately completed and observed that these events were accompanied by possible negative fluctuations in rILN activity (*Figure 3E*; time relative to head entry: F[1,12]=3.536, p=0.085; schedule: F[2,24]=0.502, p=0.61; interaction: F[2,24]=0.019).

We next assessed the generalizability of this reward-related signal in non-operant paradigms. We recorded rILN activity from mice freely moving in an arena with strawberry milk located in two corners. rILN→DS activity increased when mice approached the strawberry milk-baited corners but not the opposing non-baited arena corners (*Figure 3F*; time relative to approach: F[1,5]=12.13, p=0.018; corner: F[1,5]=12.54, p=0.017; interaction: F[1,5]=7.965, p=0.037). To directly examine rILN→DS activity relative to reward consumption, we recorded from mice drinking sucrose water from bottles connected to a lickometer. rILN→DS activity significantly increased relative to lick bout onset (*Figure 3G*; t=3.210; p=0.005).

We next tested whether reward devaluation may influence rILN→DS activity. Mice were given free access to sucrose pellets prior to 30 min test sessions in which pellets were pseudo-randomly delivered. Devaluation reduced both latency (*Figure 3H*; t=3.953, P=0.002) and frequency of pellet receptacle head entries (*Figure 3I*; t=5.068, p=0.0004) but did not significantly alter rILN→DS activity aligned to reward retrieval (*Figure 3J*; time relative to head entry: F[1,11]=12.23, p=0.003; reward value: F[1,11]=0.642, p=0.44).

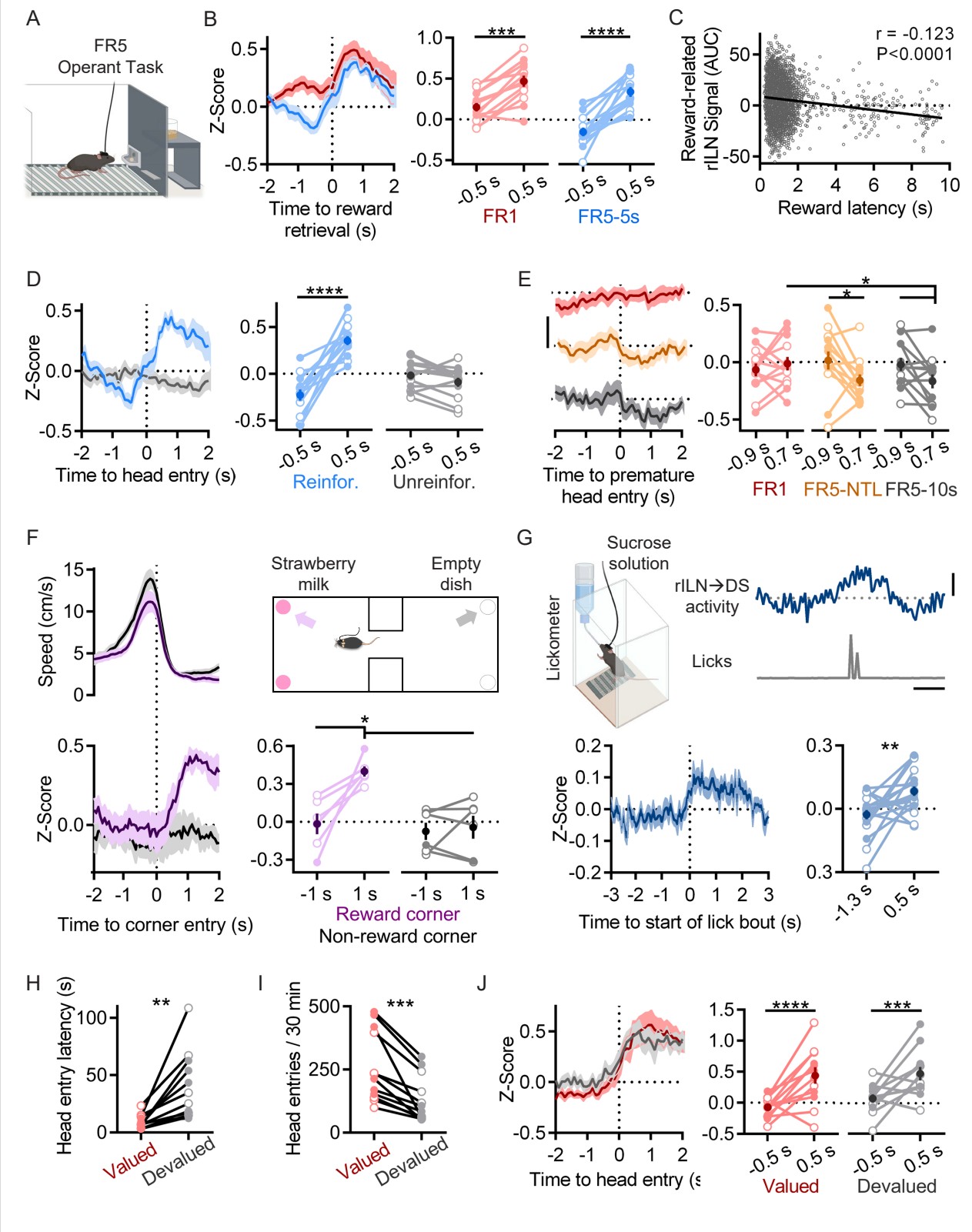

**Figure 3.** Rostral intralaminar nuclei (rILN)→dorsal striatum (DS) neurons activate with reward acquisition. (**A**) Cartoon of operant chamber. (**B**) Left: rILN→DS activity relative to sucrose pellet retrieval on FR1 (red) and FR5-5s (blue) sessions. Right: rILN→DS activity significantly increased following reward acquisition (N=13). (**C**) Across all FR5 training schedules, reward retrieval-associated rILN→DS activity negatively correlated with retrieval latency (n=6338). (**D**) Left: rILN→DS activity aligned to reward receptacle head entry following completed FR5 trials on reinforced (blue) and unreinforced (gray)

*Figure 3 continued on next page*

*Figure 3 continued*

FR5-5s sessions. Right: rILN→DS activity increased on reinforced sessions only. (**E**) Left: rILN→DS activity aligned to premature reward receptacle head entries on completed FR1 trials (red), FR5-NTL (orange), and FR5-10s (gray) schedules. Right: rILN→DS activity significantly decreased on FR5-NTL and –10 s protocols. (**F**) Movement speed (top left) and corresponding rILN→DS activity (bottom) from mice approaching strawberry milk-containing (purple) and empty (gray) corners of a two-chambered arena (top right; N=6). rILN→DS activity increased following entry to strawberry milk-baited corners (top right). (**G**) rILN→DS activity relative to sucrose water consumption. Top left: sample trace of rILN→DS activity (blue) and corresponding licks (gray). Top right: cartoon of drinking chamber. Bottom: average rILN→DS activity aligned to onset of lick bouts (left) which significantly increased following bout onset (right; N=19). (**H–J**) rILN→DS activity was recorded from food-restricted mice over multiple sessions in which sucrose pellets were pseudo-randomly dispensed (valued condition; red). Mice were then recorded for multiple sessions that were preceded by 30 min free feeding of sucrose pellets in their home cages (devalued condition; gray; N=12). (**H**) Mice retrieved dispensed sucrose pellets at a slower latency in the devalued state. (**I**) Mice entered the food receptacle fewer times during devalued sessions. (**J**) rILN→DS activity aligned to sucrose pellet retrieval was not significantly different between valued and devalued states. Scale bars: 0.3 z-score (**E**); 1 s and 0.2 z-score (**G**). Two-way repeated measures ANOVA (**B, D–F, and J**); linear correlation (**C**); paired t-test (**G, H, and I**). N=number of mice; n=number of trials. Filled and open data points represent male and female mice, respectively. Error bars = SEM. * p<0.05, ** p<0.01, *** p<0.001, and **** p<0.0001.

The online version of this article includes the following source data and figure supplement(s) for figure 3:

**Source data 1.** Source behavioral and photometric data for *Figure 3*.

**Figure supplement 1.** Rostral intralaminar nuclei (rILN)→dorsal striatum (DS) activity does not correlate with reward-predicting Pavlovian cues or changes in operant task reward probability.

**Figure supplement 1—source data 1.** Source behavioral and photometric data for *Figure 3—figure supplement 1*.

## rILN→DS neuronal activity is necessary and sufficient for optimal action performance

To causally test the role of rILN→DS activity, we optogenetically inhibited dorsal striatal projecting rILN neurons in well-trained mice performing the FR5 task (*Figure 4A*). Delivering blue light during an epoch encompassing the action initiation and reward retrieval events (*White et al., 2018*; *White et al., 2020*) pseudo-randomly on 33% of the trials (*Figure 4B*) to halorhodopsin-eYFP (NpHR-eYFP) or control eYFP-expressing mice, we found that rILN→DS neuronal inhibition resulted in fewer completed FR5 trials (*Figure 4C*; light delivery: F[1,26]=21.90, p<0.0001; virus: F[1,26]=0.390, p=0.54) and more omissions (*Figure 4D*; light: F[1,26]=13.99, p=0.0009; virus: F[1,26]=0.127, p=0.72; interaction: F[1,26]=13.10, p=0.0013; *Figure 4—figure supplement 1*). We also observed a small but significant effect of light delivery on the percentage of incomplete trials (*Figure 4E*; light: F[1,26]=4.834, p=0.037; virus: F[1,26]=2.306, p=0.141).

We previously demonstrated that optogenetic activation of rILN→DS terminals reinforces actions in an intracranial self-stimulation paradigm (*Cover et al., 2019*). However, it is unknown whether activation of this pathway at action initiation also supports the execution of rewarded action sequences. To test this, we virally expressed channelrhodopsin (ChR2-eYFP) or a control fluorophore (eYFP) in rILN→DS neurons (*Figure 4F*). Blue light was delivered on 33% of trials for 2 s starting 1 s prior to lever extension (to promote the lever press action) in trained mice (*Figure 4G*). ChR2-eYFP mice completed more FR5s on light-paired trials (*Figure 4H*; light: F[1,25]=24.39, p<0.0001; virus: F[1,25]=0.002, p=0.97; interaction: F[1,25]=13.21, p=0.0013) and, correspondingly, had fewer omitted trials (*Figure 4I*; light: F[1,25]=29.46, p<0.0001; virus: F[1,25]=0.656, p=0.43; interaction: F[1,25]=18.85, p=0.0002), compared to both non-light delivered trials and eYFP controls (*Figure 4—figure supplement 1*). We did not find significant differences in the number of incomplete FR5 trials (*Figure 4J*; light: F[1,25]=0.358, p=0.55; virus: F[1,25]=1.441, p=0.24).

## rILN→DS neurons receive subcortical and cortical inputs

Although an extensive number of inputs to the rILN are identified (*Van der Werf et al., 2002*), it is largely unknown which of these afferents may relay to the striatum to possibly contribute to function. Thus, we sought to qualitatively identify afferents impinging specifically onto rILN→DS neurons. To do this, we applied a viral strategy involving injection of nuclei that are upstream of rILN with an anterograde trans-synaptic AAV1 viral construct (AAV1-hSyn-Cre-WPRE-hGH; *Zingg et al., 2017*) that delivers cre recombinase to rILN neurons. In addition, we injected a retrogradely traveling cre-dependent tdTomato-expressing virus (AAVrg-CAG-FLEX-tdTomato-WPRE) in the DS. This viral approach produces tdTomato labeling in all rILN→DS neurons post-synaptic to the neurons at the site of AAV1-cre injection (*Figure 5—figure supplement 1*). We used this method to interrogate known

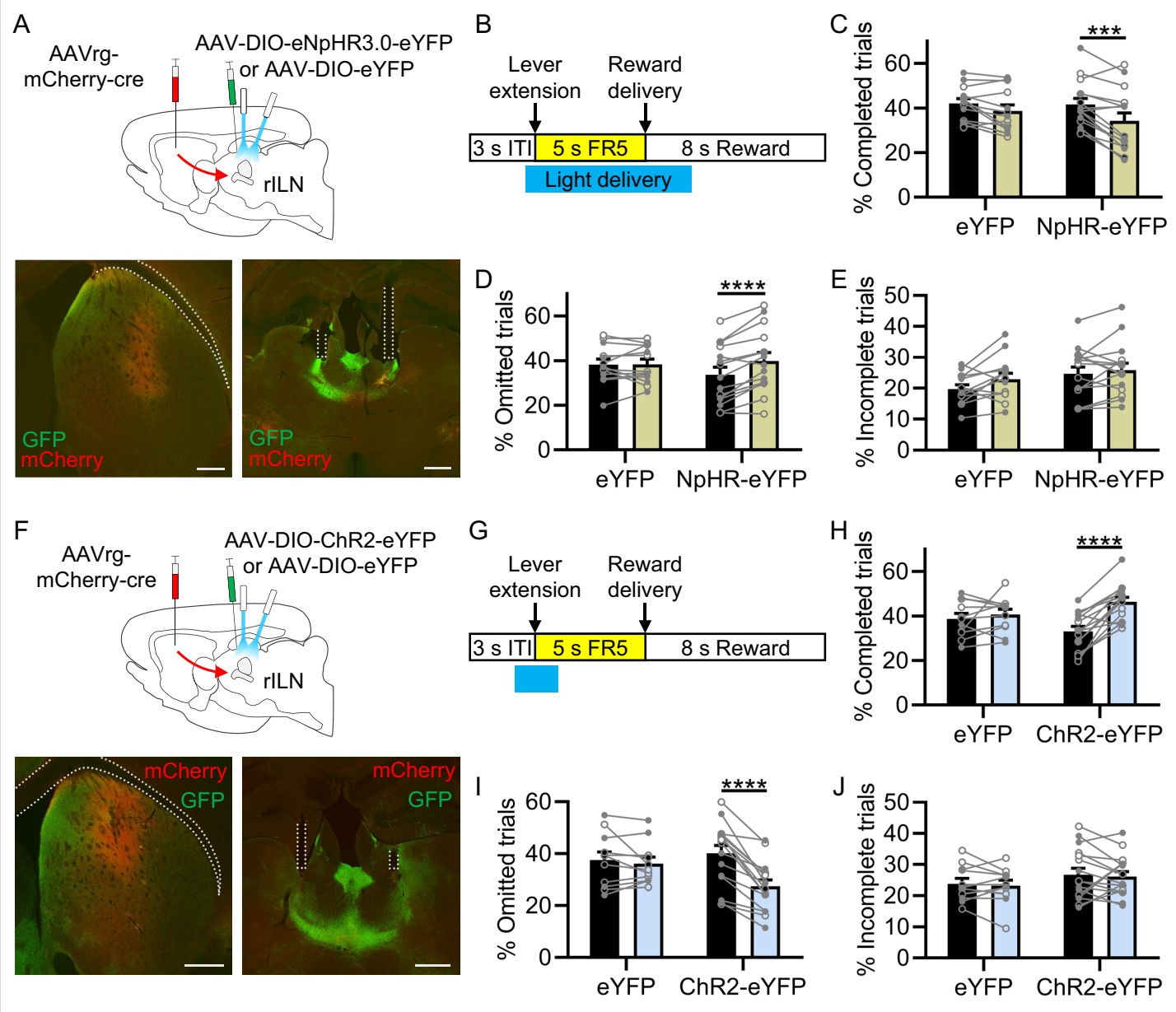

**Figure 4.** Modulating rostral intralaminar nuclei (rILN)→dorsal striatum (DS) neuronal activity bidirectionally alters FR5 performance. (**A**) Top: strategy for optical inhibition of rILN→DS neurons in FR5-trained mice. Bottom: representative cre recombinase (mCherry) and halorhodopsin (eNpHR-eYFP; GFP) expression in DS (left) and rILN (right) with optical fibers implanted in the rILN. (**B**) Schematic of 470 nm light delivery during the FR5 trial starting 0.5 s prior to lever extension and terminating either 2 s following FR5 completion or at time of lever retraction (for incomplete and omitted FR5 trials). Light was delivered pseudo-randomly on 33% of the trials. (**C**) NpHR-eYFP-expressing (N=15) but not eYFP control (N=13) mice completed fewer FR5s on light-delivered trials (yellow) than non-light-delivered trials (black). (**D**) NpHR-eYFP but not eYFP-expressing mice had a greater omission rate on light-delivered trials. (**E**) Light delivery did not alter the proportion of incomplete FR5 trials for either group. (**F**) Top: strategy for optogenetic activation of rILN→DS neurons. Bottom: representative mCherry-cre (mCherry) and channelrhodopsin (ChR2-eYFP; GFP) expression in DS (left) and rILN (right) with optic fibers implanted in the rILN. (**G**) 2 s 470 nm blue light was delivered 1 s prior to lever extension on 33% of trials. (**H**) ChR2-eYFP mice (N=16) completed more FR5s on light-delivered trials (blue) compared to non-light-delivered trials (black) and eYFP control mice (N=11). (**I**) ChR2-eYFP mice had fewer omissions on light-delivered trials than non-light-delivered trials and eYFP controls. (**J**) There were no significant differences in the rate of incomplete FR5 trials. Scale bars: 500 μm. Two-way repeated measures ANOVA (**C–E and H–J**). N=number of mice. Filled and open data points represent male and female mice, respectively. Error bars = SEM. *** p<0.001 and **** p<0.0001. See *Figure 4—figure supplement 1* for summary statistics.

The online version of this article includes the following source data and figure supplement(s) for figure 4:

**Source data 1.** Source behavioral data for *Figure 4*.

**Figure supplement 1.** Rostral intralaminar nuclei (rILN)→dorsal striatum (DS) manipulation alters FR5 task performance.

excitatory inputs to the rILN. We found that both the anterior cingulate cortex (ACC; *Figure 5A*) and orbitofrontal cortex (OFC; *Figure 5B*) synapse on ipsilateral rILN→DS neurons and lesser so on contralateral rILN→DS neurons. We also found that projections arising from the glutamatergic/GABAergic hypothalamic supramammillary and lateral (S/L) nuclei (*Hashimotodani et al., 2018*; *Stuber and Wise, 2016*) innervate the rILN→DS neurons residing in CM and contralateral CL (*Figure 5C*). Previous tracing studies indicate that the rILN receive input from the basal ganglia output projection, the substantia nigra (SN) pars reticulata (*Kaufman and Rosenquist, 1985*), as well as from other targets of nigral output including the superior colliculus (SC; *Yamasaki et al., 1986*), reticular formation (RF; *Krout et al., 2002*), and pedunculopontine nucleus (PPN; *Hallanger et al., 1987*; *Huerta-Ocampo et al., 2020*). We observed that projections from all four of these regions synapse on rILN→DS neurons (*Figure 5D–G*). These neuronal tract tracing results are quantified (*Figure 5—figure supplement 2*).

## Discussion

Our results demonstrate that rILN→DS neurons stably activate at action initiation and reward acquisition and that manipulations of this circuit significantly impact action execution. We found that rILN→DS pathway activation biased mice to initiate more rewarded lever presses, whereas inhibitory manipulations resulted in fewer completed press sequences. These results complement our previous findings of rILN suppression reducing overall movement (*Cover et al., 2019*). Together, these findings implicate the rILN→DS pathway as a critical contributor to action reinforcement and, therefore, performance. We also observed that rILN→DS projection neurons exhibit homogeneous physiological properties across the three rILN, which display relative differences in the afferents impinging upon them. How these differences culminate in unique functional features across these nuclei requires further study.

Corticostriatal circuits guide action learning and habit formation (*Kupferschmidt et al., 2017*; *Smith and Graybiel, 2013*; *Yin and Knowlton, 2006*) by coordinating the activity of striatal output neurons (*Tecuapetla et al., 2016*; *Yin et al., 2009*). Thalamic inputs are suggested to influence action expression through modulation of these corticostriatal circuits (*Ding et al., 2010*). However, rILN signaling also directly drives striatal medium spiny neuron (MSN) activity (*Chen et al., 2014*; *Ellender et al., 2013*) and cholinergic interneuron firing (*Cover et al., 2019*). Our observation of rILN→DS neuronal activity occurring at action initiation and correlating to initiation latency suggests that this excitatory input may directly drive striatal signaling for action execution. The absence of discrete signal changes for presses two through five in mice that are well-trained on the FR5 task may reflect the concatenation of the individual movements into a fluid sequence (*Jin et al., 2014*). Alternatively, press-related rILN→DS neuronal activity may be obscured due to the temporal limits of the calcium sensor and photometric system. Regardless, these results support previous observations of rILN activity mediating motor control (*Chen et al., 2014*; *Giber et al., 2015*; *Luma et al., 2022*) and action switching (*Kato et al., 2018*).

Our finding that rILN→DS neurons activate at reward acquisition presents, to our knowledge, a unique addition to known reward-related circuitry. Midbrain dopamine neuron firing shifts during reinforcement learning from reward presentation to associated cues and unexpected reward presentation or omission to provide teachable reward prediction errors (*Schultz, 1998*). In contrast, we found that the rILN stably and persistently signals at reward acquisition across all stages of training and in multiple contexts. Similar reward-related activity is observed in the rILN of primates performing an oculomotor task (*Wyder et al., 2003*). This signal may serve as a mechanism to provide ongoing reinforcement of appropriately selected actions. Such a mechanism would be advantageous for both the learning of rewarded action sequences but also provides a salient omission when action outcomes change. The sudden absence of this consistent signal may provide a simple cue to drive the search for a new action plan that results in reward. Indeed, we observed a modest negative signal fluctuation when mice prematurely check for sucrose pellets mid-FR5 sequence suggesting that a potential bidirectional signal may provide an instructive cue for action reinforcement. Inhibition of rILN→DS activity significantly reduced the execution of FR5 pressing, which may be due to degradation of this putative reinforcement signal. Regardless, the fiber photometry findings should be considered cautiously given that this approach may not sensitively detect the activity of minority neuronal populations.

Dopamine release ramps with proximity to rewards and scales to reward magnitude (*Hamid et al., 2016*; *Howe et al., 2013*; *Mohebi et al., 2019*), in addition to dopamine terminal signaling coinciding with reward presentation (*Howe and Dombeck, 2016*). Through a cholinergic intermediary, the rILN

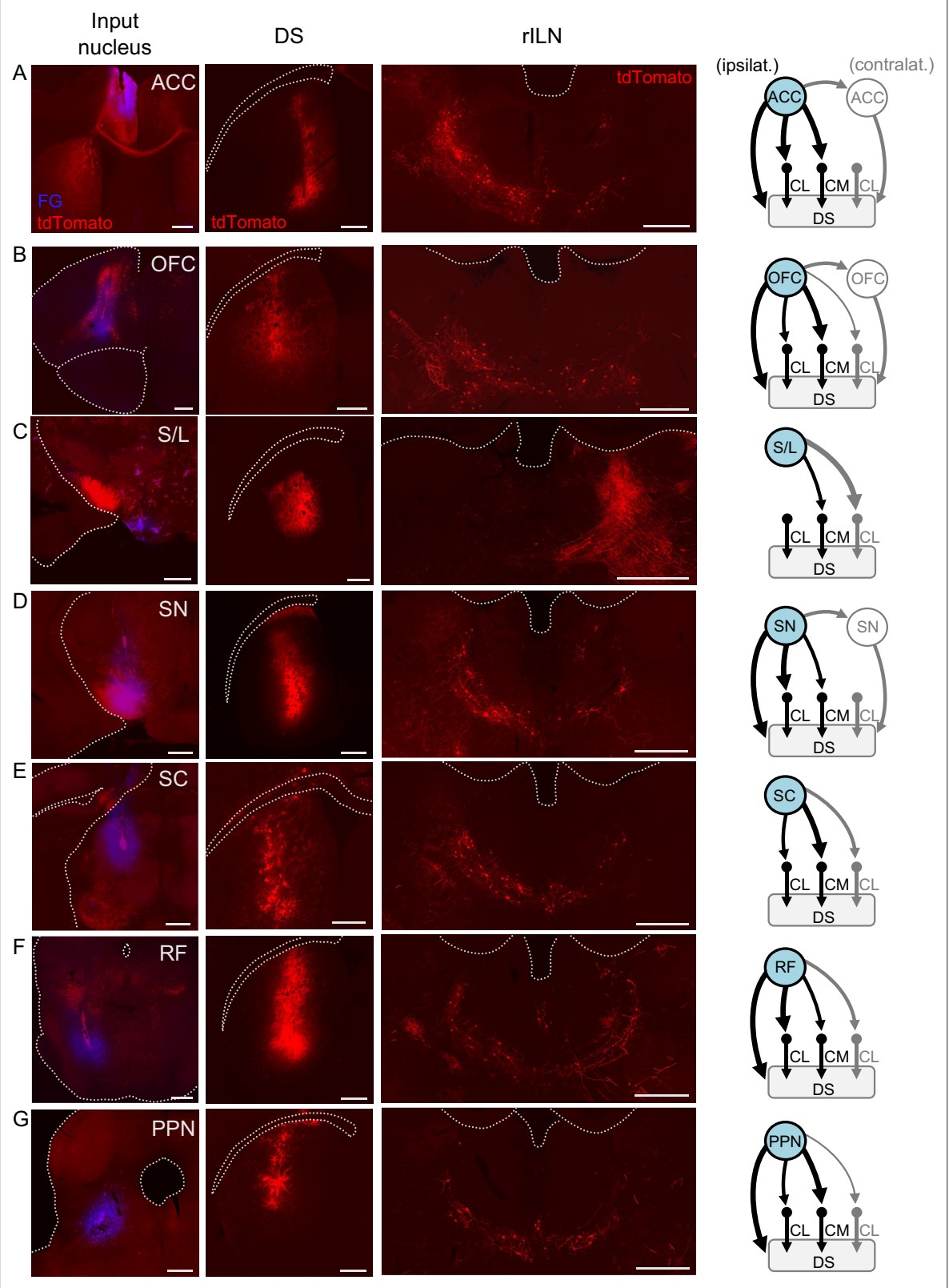

**Figure 5.** Afferent innervation of rostral intralaminar nuclei (rILN)→dorsal striatum (DS) neurons. (**A**) Left: injection sites of AAV1-cre and fluorogold (FG) in the anterior cingulate cortex (ACC). Middle left: injection site for retrograde cre-dependent tdTomato expressing virus in the central DS. Middle right: representative tdTomato labeling of DS-projecting rILN neurons post-synaptic to ACC projections. Right: summary of ACC connectivity to rILN nuclei, with the blue filled nuclei indicating the injected hemisphere. (**B–G**) Same as **A** but for orbitofrontal cortex (OFC; **B**), supramammillary and lateral

*Figure 5 continued on next page*

*Figure 5 continued*

hypothalamus (S/L; **C**), substantia nigra (SN; **D**), superior colliculus (SC; **E**), reticular formation (RF; **F**), and pedunculopontine nucleus (PPN; **G**). Summary diagrams are based on two to four cases per region. Line thickness indicates a difference of ≥40% of labeled neurons between the compared rILN nuclei and (if applicable) contralateral versus ipsilateral direct projections to the striatum. Gray lines indicate circuits contralateral to the injected hemisphere. See *Figure 5—figure supplement 2* for circuit quantification. Scale bars: 500 μm.

The online version of this article includes the following figure supplement(s) for figure 5:

**Figure supplement 1.** Trans-synaptic neuronal tract tracing to identify specific afferents to the rostral intralaminar nuclei (rILN)→dorsal striatum (DS) circuit.

**Figure supplement 2.** Differences in afferent connectivity to rostral intralaminar nuclei (rILN)→dorsal striatum (DS) projection neurons across the three rILN nuclei.

are capable of locally inducing striatal dopamine release in a behaviorally significant manner (*Cover et al., 2019*). Our finding of rILN signaling at reward acquisition may suggest that this local dopamine release mechanism is evoked at such events, which may supplement somatically driven dopamine signaling (*Cohen et al., 2012*). The rILN may also relay reward-related information directly to MSNs as these cells are shown to encode reward value and action outcome (*Hori et al., 2009*; *Lauwereyns et al., 2002*; *Nonomura et al., 2018*). The present study photometrically recorded rILN somatodendritic activity and may not fully reflect signaling at the level of the axon terminal (*Legaria et al., 2022*), including calcium-dependent presynaptic plastic changes occurring at these terminals in the striatum. Future work investigating rILN axonal signals in the striatum may reveal further complexity in thalamic activity supporting action learning and motivated behavior.

One question that emerges from these results is the origin of the rILN→DS action initiation and reward-related signals. Our neuronal tract tracing experiments identified a range of afferents impinging on rILN→DS neurons. The ACC and OFC, which themselves receive input from the rILN (*Hunnicutt et al., 2014*; *Murphy and Deutch, 2018*; *Van der Werf et al., 2002*), provide two sources of excitatory input. These cortical regions are broadly associated with decision making. The OFC encodes outcome value (*Gremel and Costa, 2013*; *Malvaez et al., 2019*; *Stolyarova and Izquierdo, 2017*; *Zhou et al., 2019*), whereas the ACC encodes decision predictions, surprise signals, and prediction errors, in addition to mediating cognitive control (*Hayden et al., 2011*; *Shenhav et al., 2013*; *Totah et al., 2009*; *Wallis and Kennerley, 2011*). Although less studied, the S/L hypothalamus are well connected with mesolimbic circuitry and linked to regulation of feeding behavior (*Plaisier et al., 2020*; *Stuber and Wise, 2016*). Together, these afferents may impart value and motivation for rILN→DS guided action selection.

We also determined that rILN→DS neurons receive input from a basal ganglia output center, the SN pars reticulata, as well as from nuclei that themselves receive input from the SN pars reticulata: the SC, RF, and PPN. These results confirm the presence of both direct and indirect basal ganglia subcortical loops relaying back to the DS (*Alexander et al., 1986*). Through inhibition driven by the SN and selective disinhibition enabled by D1-MSN pathway signaling, these loops are hypothesized to support the main functions of the basal ganglia: action selection and reinforcement learning (*McHaffie et al., 2005*; *Redgrave et al., 2011*). The rILN are uniquely positioned to provide feedback for these functions, and our observation of rILN activity corresponding to action initiation and reward acquisition supports this notion. Taken together, rILN pathology may contribute to disorders of action engagement: rILN hyperactivity may contribute to drug seeking (*Li et al., 2018*; *Wang et al., 2019*) in a role similar to the adjacent paraventricular thalamus (*Hamlin et al., 2009*; *Matzeu et al., 2015*), or to attention deficit hyperactivity disorder (*Jones et al., 2020*), while hypoactivity may contribute to impaired goal-directed behavior in major depression (*Höflich et al., 2019*).

The neuronal tract tracing experiments revealed specificity of some afferents targeting either the CL or CM nuclei. Such differences in anatomical connectivity may confer individual rILN with selective activity during reward-driven action sequences. As our approach to in vivo rILN recording and manipulation prevented us from parsing the contributions of the individual nuclei, we could not determine whether the action initiation and reward acquisition signals are uniformly represented across the three rILN. Future studies targeting individual rILN circuits may identify specific contributions to motivated behavior.

# Materials and methods

**Key resources table**

| Reagent type (species) or resource | Designation | Source or reference | Identifiers | Additional information |
|---|---|---|---|---|
| Strain and strain background (*Mus musculus*) | Wild-type | C57BL/6J; Jackson Laboratory | #000664 | Male and female; 2–5 months old |
| Antibody | Goat polyclonal anti-GFP | Abcam | #ab6673 | 1:2000 |
| Antibody | Chicken polyclonal anti-mCherry | Novus Biologicals | #NBP2-25158 | 1:2000 |
| Antibody | Donkey polyclonal anti-goat conjugated to Alexa Fluor 488 | Jackson ImmunoResearch | #703-545-155 | 1:1500 |
| Antibody | Donkey polyclonal anti-chicken conjugated to Alexa Fluor 594 | Jackson ImmunoResearch | #703-585-155 | 1:1500 |
| Software and algorithm | Clampex 10.4.1.4 | Molecular Devices | | |
| Software and algorithm | Clampfit 10.4.1.4 | Molecular Devices | | |
| Software and algorithm | Med-PC | Med Associates Inc. | | |
| Software and algorithm | Ethovision XT | Noldus | | |
| Software and algorithm | Prism 6.01 | Graphpad | | |
| Software and algorithm | Matlab R2019a | The MathWorks Inc. | | |
| Other (AAV) | AAVrg-EF1a-mcherry-IRES-Cre | Addgene | #55632-AAVrg | See Materials and methods: Surgical procedures |
| Other (AAV) | AAV5-Syn-Flex-GCaMP6s | Addgene | #100845-AAV5 | See Materials and methods: Surgical procedures |
| Other (AAV) | AAV5-EF1a-DIO-eNpHR3.0-eYFP-WPRE | Addgene | #26966-AAV5 | See Materials and methods: Surgical procedures |
| Other (AAV) | AAV5-EF1a-DIO-hSyn-ChR2-eYFP-WPRE | Addgene | #20298-AAV5 | See Materials and methods: Surgical procedures |
| Other (AAV) | AAV5-EF1a-DIO-eYFP-WPRE | Addgene | #27056-AAV5 | See Materials and methods: Surgical procedures |
| Other (AAV) | AAVrg-CAG-tdTomato | Addgene | #59462-AAVrg | See Materials and methods: Surgical procedures |
| Other (AAV) | AAVrg-CAG-FLEX-tdTomato-WPRE | Addgene | #28306-AAVrg | See Materials and methods: Surgical procedures |
| Other (AAV) | AAV1-hSyn-Cre-WPRE-hGH | Addgene | #105553-AAV1 | See Materials and methods: Surgical procedures |
| Other (motivator) | Sucrose pellet, 14 mg | BioServ | #F05684 | See Materials and methods: Behavioral testing |

## Subjects

Two- to five-month-old male and female C57BL/6J (wild-type; Jackson Laboratory, #000664) mice were housed in a temperature and humidity controlled vivarium under a 12 hour light/dark cycle (lights on at 0700 hours). Mice were housed with littermates (two to five) per cage, except for those singly housed following fiber or cannula implantation. Mice performing operant tasks were weighed and fed daily to maintain 90% of ad libitum weight; all others received ad libitum food and water. All experiments were performed in accordance with the United States Public Health Service Guide for Care and Use of Laboratory Animals and were approved by the Institutional Animal Care and Use Committee (AUP 0522009) at the University of Maryland, Baltimore. Sample sizes were determined based on

publication standards. Studies were completed across multiple cohorts that produced homogeneous experimental findings. Animals were randomly assigned to control and experimental groups. Group assignment was also balanced to ensure equal numbers of males and females between experimental and control cohorts. Blinding was not used in the present study as all subjects and data were treated with identical equipment, analysis methods, and exclusion criteria.

## Surgical procedures

Mice were anesthetized with isoflurane (5% induction, 1–2% maintenance) and head-fixed in a mouse stereotaxic apparatus (Kopf Instruments, Tujunga, CA, USA). Bupivacaine hydrochloride (0.25%; s.c.) was applied to the scalp prior to incision. Viruses were infused at a rate of 25 nl/min using a 25 G syringe (Hamilton Company, Reno, NV, USA). All mice received carprofen (5 mg/kg; s.c.) and recovered for a minimum of 7 days to prior to behavioral testing.

For fiber photometry experiments, AAVrg-EF1a-mcherry-IRES-Cre (Addgene #55632-AAVrg) was unilaterally injected in the central DS (600 nl; distance from bregma in mm, [anterior-posterior, AP] +0.65, [medial-lateral, ML] 1.70, and [dorsal-ventral, DV] –3.00), and two unilateral injections of AAV5-Syn-Flex-GCaMP6s (Addgene #100845-AAV5) were made in the rILN (300 nl/injection; [AP] –1.25 and –1.95, [ML] 0.75, and [DV] –3.00). In a separate surgery 3–4 weeks after, a low numerical aperture (NA; 0.22) fiber (200 um core, Thorlabs, Newton, NJ, USA) was implanted in the rILN ([AP] –1.60, [ML] 0.76, and [DV] –3.00). For optogenetic manipulations, AAVrg-EF1a-mCherry-IRES-Cre was bilaterally injected in the DS (as stated above), and either AAV5-EF1a-DIO-eNpHR3.0-eYFP-WPRE (Addgene #26966-AAV5), AAV5-EF1a-DIO-hSyn-ChR2-eYFP-WPRE (Addgene #20298-AAV5), or AAV5-EF1a-DIO-eYFP-WPRE (Addgene #27056-AAV5) was injected in the rILN (350 nl/hemisphere; [AP] –1.40, [ML] ±0.75, and [DV] –3.50). Two optic fibers (NA 0.66, 200 μm core, Prizmatix, Givat-Shmuel, Israel) were later implanted in the rILN: one at ([AP] –1.60, [ML] –0.75, and [DV] –2.70) and the second at a 20° AP angle ([AP] –2.58, [ML] +0.75, and [DV] –3.03). All implants were secured with skull screws (BASi, West Lafayette, IN, USA) and dental cement.

For electrophysiology experiments, retrograde AAVrg-CAG-tdTomato (Addgene #59462-AAVrg) was injected in the DS (same coordinates and volume as previously listed) in wild-type mice to patch tdTomato-labeled rILN neurons.

Trans-synaptic tract tracing was performed to identify inputs to rILN→DS neurons. Mice were bilaterally injected with AAVrg-CAG-FLEX-tdTomato-WPRE in the DS (Addgene #28306-AAVrg; 600 nl/hemisphere; [AP] +0.65, [ML] ±1.70, and [DV] –3.00). Afferent regions of interest were unilaterally injected with AAV1-hSyn-Cre-WPRE-hGH (Addgene #105553-AAV1) as well as Fluorogold (20 nl; Fluorochrome LLC, Denver, CO, USA) to label the injection site. The unilateral injection hemisphere was counterbalanced across animals for each input region. Stereotaxic coordinates and virus volumes are as follows: OFC ([AP] +2.50, [ML] 1.00, and [DV] –2.20; 250 nl), ACC ([AP] +1.00, [ML] 0.30, and [DV] –1.10; 200 nl), S/L ([AP] –2.40, [ML] 0.50, and [DV] –5.65; 150 nl), SN ([AP] –3.20, [ML] 1.50, and [DV] –4.80; 250 nl), SC ([AP] –3.50, [ML] 0.80, and [DV] –2.75; 240 nl), RF ([AP] –4.20, [ML] 0.75, and [DV] –4.25; 310 nl), and PPN ([AP] –4.70, [ML] 1.20, and [DV] –3.00; 250 nl).

Immunohistochemistry was performed to confirm virus expression and fiber placement; animals were excluded from experiments for poor virus expression at target regions or viral expression in non-target regions (i.e. the caudal intralaminar parafascicular nucleus).

## Immunohistochemistry

Mice were transcardially perfused with room-temperature 0.1 M PBS, pH 7.3, followed by ice cold 4% (wt/vol) paraformaldehyde in PBS. Brains were extracted and post-fixed with 4% paraformaldehyde in PBS at 4°C. 50 μm coronal sections were cut using a Leica VT100S vibrating microtome. Goat anti-GFP (Abcam #ab6673, Waltham, MA, USA) and chicken anti-mCherry (Novus Biologicals #NBP2-25158, Littleton, CO, USA) primary antibodies were used at a 1:2000 dilution to amplify GCaMP6s and cre expression, respectively. Chicken anti-mCherry was also used to amplify virally expressed tdTomato. Donkey anti-goat conjugated to Alexa Fluor 488 (Jackson ImmunoResearch #703-545-155, West Grove, PA, USA) and donkey anti-chicken conjugated to Alexa Fluor 594 (Jackson ImmunoResearch #703-585-155) secondary antibodies were used at a 1:1500 dilution. The Brain BLAQ protocol (*Kupferschmidt et al., 2015*) was used for immunohistochemistry following electrophysiology.

## Slice electrophysiology

Mice were deeply anesthetized with isoflurane before rapid decapitation and brain removal. 250 µm thick coronal slices were prepared in ice cold, 95% oxygen, 5% carbon dioxide (carbogen)-bubbled modified artificial cerebrospinal fluid (aCSF) (In mM: 194 sucrose, 30 NaCl, 4.5 KCl, 1 MgCl2, 26 NaHCO3, 1.2 NaH2PO4, and 10 D-glucose) before incubating at 32°C for 30 min in aCSF (in mM: 124 NaCl, 4.5 KCl, 2 CaCl2, 1 MgCl2, 26 NaHCO3, 1.2 NaH2PO4, and 10 D-glucose). Slices were then stored at room temperature until recording.

Thalamic brain slices were transferred to the recording chamber and perfused with carbogen-bubbled aCSF (in mM; 124 NaCl, 4.5 KCl, 2 CaCl2, 1 MgCl2, 26 NaHCO3, 1.2 NaH2PO4, and 10 D-glucose) at 29–31°C. Whole-cell recordings were made with borosilicate glass pipettes (2–5 MΩ). aCSF was perfused on slices through a gravity perfusion system. DS projecting rILN neurons were recorded with a potassium-based internal solution (in mM: 126 potassium gluconate, 4 KCl, 10 HEPES, 4 ATP-Mg, 0.3 GTP-Na, and 10 phosphocreatine; 290–295 mOsm; pH 7.3) containing a hydrazide dye conjugated with Alexa Fluor 488 (40 mM) to allow for post-hoc identification of cell location within the rILN. Membrane capacitance and resistance were collected at –60 mV voltage clamp configuration, whereas the resting membrane potential was obtained in current-clamp with a MultiClamp 700B amplifier (Molecular Devices, San Jose, CA, USA); recordings were filtered and digitized at 2 and 10 kHz, respectively. Input resistance was calculated from the mV change induced by a 140 pA hyperpolarizing current step. Data was acquired with Clampex 10.4.1.4 and analyzed with Clampfit 10.4.1.4 (Molecular Devices). Calculation of the accommodation index for rILN AP spiking was based on the entire 0.5 s current injection step that elicited maximum firing as previously calculated (*White and Mathur, 2018*). Cells that did not maintain sustained firing (i.e. resulted in a depolarization block) were excluded from this analysis. Cells were determined to reside in CL, PC, and CM nuclei based on the *Franklin and Paxinos, 2008*.

## Behavioral testing

### FR5 lever press operant paradigm

Food-restricted mice were trained to lever press for pellets in an operant chamber (21.6 × 17.8 × 12.7 cm chamber, Med Associates Inc, Fairfax, VT, USA) containing a retractable lever and a trough pellet receptacle equipped with an infrared beam that delivered 14 mg sucrose pellets (#F05684, BioServ, Frenchtown, NJ, USA). On each trial, the lever extended into the chamber and retracted once the mouse retrieved a rewarded sucrose pellet (for completed trials) or after the response time limit passes. Mice received two 30 min training sessions per day with criterion to progress to the next training schedule consisting of ≥30 completed fixed-ratio trials for two consecutive sessions. Training started on FR1 and progressed to FR3 and FR5, all without time limit restrictions to complete the press sequence. Mice were then trained to complete the FR5 sequence under increasing time constraints (time from lever extension), progressing through 30 s, 15 s, 10 s, 7.5 s, and 5 s schedules. Some mice were unable to successfully perform on FR5-5s and remained at FR5-7.5s for further testing. Optogenetic manipulations were administered once mice achieved consistent performance at or above criterion on their terminal protocol (approximately 8–10 sessions).

A variation of the FR5 lever press task was conducted in which sessions alternated between sucrose pellet reinforcement rates of 100 and (pseudo-randomly) 50% (*Figure 3—figure supplement 1* H–I). Fiber photometric data was analyzed for all completed FR5 trials following completion of five FR5s (for 100% reinforcement sessions) and five completed but unreinforced FR5s (50% reinforcement sessions) per session.

### In vivo optogenetics

For FR5 optogenetic experiments, 470 nm light was delivered bilaterally during experimental sessions using an LED system (Plexon; Dallas, TX, USA). Our group previously demonstrated that 470 nm light sufficiently activates eNpHR channels to inhibit neuronal activity (*White et al., 2018*; *White et al., 2020*). For optical inhibition, light was delivered pseudo-randomly on 33% of the trials with performance averaged across four consecutive sessions, comparing performance with non-light delivered trials. For optical activation, light was delivered pseudo-randomly on 33% of the trials with performance averaged across six consecutive sessions, comparing performance with non-light delivered trials. For both experiments, mice with the highest and lowest difference in performance were

excluded from analysis (both experimental and control cohorts). Experimenters were not blinded to animal group designation.

## Pavlovian appetitive conditioning

To assess for rILN→DS neuronal activity related to reward-related cues, food-restricted behaviorally naïve mice expressing GCaMP6s in rILN→DS neurons with a multimode fiber implanted in the rILN underwent a Pavlovian conditioned reinforcement paradigm and were randomly assigned to one of two cohorts. For one cohort ('tone-paired'), a 2 s 10 kHz tone was presented pseudo-randomly during 30 s long trials and co-terminated with the dispensing of a sucrose pellet. A control cohort ('tone-unpaired') received tone and pellet each pseudo-randomly. Mice received six 30 min sessions of these respective paradigms. To test for the effect of reward devaluation on rILN→DS neuronal activity, mice received an additional three sessions of their respective protocols that were each preceded by 30 min of free feeding of sucrose pellets in their home cage.

## Movement and reward analysis

Mice expressing GCaMP6s in rILN→DS neurons were assessed for movement-related rILN activity in a two-chambered arena (70 × 30 × 25 cm) using Ethovision XT video recording (Noldus, Wageningen, the Netherlands). Following habituation and multiple 15 min recorded sessions, weigh boats filled with strawberry milk (Nesquik) were placed in two corners of the arena, and empty weigh boats were placed in the opposite two corners. rILN→DS photometric activity was aligned to when the center of the mouse body passed within 5 cm of the weigh boats.

## Sucrose consumption assay

To directly correlate rILN→DS photometric activity to reward consumption, mice were placed under a reverse 12 hour light cycle (lights off at 0900) for 2 weeks before rILN photometry recordings were collected while given access to 2 and 8% (wt/vol) sucrose water connected to a custom lickometer (*Patton et al., 2021*). Licks were recorded using Axoscope software (Molecular Devices), and rILN→DS photometry signal was aligned via custom MATLAB code to the start of lick bouts. Bouts were defined as two or more licks with an inter-lick interval of less than 2 s.

## Fiber photometry

Photometry data were collected using a customized in vivo fiber photometry system. Two single-wavelength laser modules were used: a 473 nm laser for optimal GCaMP6s excitation and a 405 nm laser to excite GCaMP6s at its isosbestic wavelength (Opto Engine, Midvale, UT, USA). Emission from 405 nm excitation was used to control for signal artifacts due to photometry cable motion, background fluorescence, and other sources of noise (*Kim et al., 2016*). The two lasers were multiplexed at 10 or 15 Hz, resulting in a continuous 20 or 30 Hz pulse train. Both laser beams were bounced into a dichroic filter cube designed for 473 and 405 nm excitation as well as for 510 nm emission (Chroma Technology Corp., Bellow Falls, VT, USA). The two excitation wavelengths were focused through a ×4 fluorite objective (Olympus, Tokyo, Japan) onto a multimode fiber bundle (Thorlabs). One fiber was connected to the mouse through a chronic unilateral multimode fiber implant, and another fiber was placed inside a tube of Alexa Fluor 488 to control for variability in laser energy. Emissions from GCaMP6s and Alexa Fluor 488 were detected as an image of the fiber bundle using an ORCA-Flash4.0 LT high-resolution CMOS camera (Hamamatsu Photonics, Hamamatsu City, Japan). Laser multiplexing and image acquisition were synchronized using an Arduino Leonardo microcontroller. Trial or time-dependent recordings were initiated through MedPC or Ethovision systems. Camera image acquisition parameters were controlled through HCImage software for Hamamatsu cameras.

## Quantification and statistical analysis

Electrophysiological data were analyzed using Clampex software. Statistical analyses were performed in Prism (version 6.01; GraphPad Software, San Diego, CA, USA) or MATLAB (R2019a; The MathWorks, Inc, Natick, MA, USA).

Photometry data were analyzed using a combination of custom MATLAB code (see *Source code 1*) and Prism. Pixel intensities imaged from the fiber implanted in the rILN and the fluorophore control fiber were first averaged. The background signal in the absence of laser transmission was

then subtracted from the averaged signals. Two separate regressions were performed to minimize any noise sources. First, 473 and 405 nm photometry signals were regressed with the corresponding control fluorophore signal, and the residuals of the regression were then used for further processing. The 473 nm signal was then regressed with the 405 nm signal as a covariate, and the residuals of the regression were extracted as the fully processed photometry signal (z-score) from the rILN (*White et al., 2020*). The photometry signal was then statistically analyzed based on performance or operant task event (i.e. movement-aligned activity or FR trial outcome). For each analysis comparing two or more categories of rILN signal, the number of averaged events or trials were standardized across each compared category for each animal. The average z-score for three consecutive time points was statistically analyzed with the appropriate t-test or ANOVA. Signal AUC was computed in MATLAB.

To quantify relative strength of afferent innervation to rILN→DS neurons identified in the di-synaptic circuit tracing study, tdTomato-labeled cells were counted in CM, ipsilateral CL, and contralateral CL. Separate ratios of cell counts for CM: ipsilateral CL and ipsilateral CL: contralateral CL were derived, correcting for area, within each coronal slice and averaged across slices and cases for a given assessed afferent. Ratios indicating a difference ≥40% of labeled neurons between the compared rILN nuclei are denoted by line thickness in *Figure 5* summary diagrams. This criterion was also used to assess relative strength of ipsilateral and contralateral afferent→afferent→DS di-synaptic circuits, when applicable.

All statistical analyses are reported in the Results. The specific statistical test as well as value and description of n are listed in the figure legends. Averaged data is expressed as mean ± SE. Correlations are expressed as Pearson's r and 95% CIs. All statistical tests were conducted two-sided, when applicable. Significant post-hoc Holm-Ŝidàk tests for ANOVAs are indicated in the figures. All asterisks in figures indicate: * $p<0.05$, ** $p<0.01$, *** $p<0.001$, and **** $p<0.0001$.

## Acknowledgements

The authors thank Dr. Alexandros Poulopoulos for microscopy assistance and Dr. Donna Calu for Pavlovian behavior consultation. https://www.BioRender.com was used to create some of the figure schematics. This work was supported by the National Institute on Alcohol Abuse and Alcoholism grant R01AA028070 (to BNM) and National Institute of Drug Abuse grant F31DA047014 (to KKC). The content is solely the responsibility of the authors and does not necessarily represent the official views of the National Institutes of Health.

## Additional information

### Funding

| Funder | Grant reference number | Author |
| --- | --- | --- |
| National Institute on Alcohol Abuse and Alcoholism | R01AA028070 | Brian N Mathur |
| National Institute on Drug Abuse | F31DA047014 | Kara K Cover |

The funders had no role in study design, data collection and interpretation, or the decision to submit the work for publication.

### Author contributions

Kara K Cover, Conceptualization, Data curation, Formal analysis, Supervision, Funding acquisition, Investigation, Writing – original draft, Writing – review and editing; Abby G Lieberman, Data curation, Formal analysis, Investigation; Morgan M Heckman, Data curation, Formal analysis; Brian N Mathur, Conceptualization, Resources, Supervision, Funding acquisition, Project administration, Writing – review and editing

### Author ORCIDs

Kara K Cover ⓘ http://orcid.org/0000-0003-3938-9669

Brian N Mathur  http://orcid.org/0000-0003-2912-8625

### Ethics

All experiments were performed in accordance with the United States Public Health Service Guide for Care and Use of Laboratory Animals and were approved by the Institutional Animal Care and Use Committee (AUP 0522009) at the University of Maryland, Baltimore.

### Decision letter and Author response

Decision letter https://doi.org/10.7554/eLife.83627.sa1
Author response https://doi.org/10.7554/eLife.83627.sa2

## Additional files

### Supplementary files

• MDAR checklist

• Source code 1. Source code for the processing and alignment of the photometric signal to experiment-related events.

### Data availability

All data generated or analysed during this study are included in the manuscript and supporting files; Source Data files have been provided for all Figures. A Source Code file is provided for photometry analysis.

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
