## [Editor Report]

Cover et al. examine the pathway from the intralaminar nucleus of the thalamus (rILN) to the dorsal striatum (DS) in the reinforcement of behavior/actions. The rILN sends a large glutamatergic projection to the DS, but its role in action selection was unknown. The authors found that the rILN neurons that project to the DS were activated at both action initiation and with the reward, and that activation and inhibition of this pathway increased the success or decreased the success of reward acquisition, respectively. The findings are an important advance in our understanding of the function of rILN to DS projection in reward-based behavior, and the manuscript has provided convincing evidence with the appropriate methodologies to support these claims.

---

## [Decision Letter]

**Decision letter after peer review:**

Thank you for submitting your article "The rostral intralaminar nuclear complex of the thalamus supports striatally-mediated action reinforcement" for consideration by *eLife*. Your article has been reviewed by 3 peer reviewers, one of whom is a member of our Board of Reviewing Editors, and the evaluation has been overseen by Kate Wassum as the Senior Editor. The reviewers have opted to remain anonymous.

The reviewers have discussed their reviews with one another, and the Reviewing Editor has drafted this to help you prepare a revised submission. Along with your revision please provide a point x point response to each reviewer critique, including the public review and recommendations for authors, play special attention to the essential revisions below.

Essential revisions:

1) The reviewers pointed out that a few strong claims are not well-supported by the experimental data, in addition, a couple of previously published studies are not accurately cited/interpreted in the manuscript. Please tone down the related statements and accurately cite the previous findings.

2) The reviewers requested further clarification, esp. the anatomical tracing part of the experimental data. For example, providing cell counts, clarification on the methods, etc. Please better integrate this part of the finding into the rest of the manuscript. Please follow the more detailed suggestions in the long-form reviews on other suggestions.

Please include a key resource table if you haven't already done so.

*Reviewer #1 (Recommendations for the authors):*

While overall this is a well-executed study that uses a variety of behavior tasks to demonstrate the activity of DS-projecting rILN neurons. Yet, the different parts of this study (electrophysiology, behavior, tracing) feel somewhat disjointed, and additional control experiments would be needed to substantiate some of the claims made.

– The statement that the authors "discovered that rILN neurons projecting to the DS integrated information from several subcortical and cortical sources" (abstract lines 39 – 41) is misleading and not fully supported by the presented data. While the authors demonstrate that rILN neurons can receive input projections from several subcortical and cortical areas, they do not show whether these projections converge onto the same individual neurons in rILN, and more importantly, they do not show whether "information" is carried by these projections or integrated into rILN neurons during behavior. Recordings and manipulation experiments of these subcortical and cortical inputs to rILN would be necessary to support such a claim.

– While fiber photometry is a powerful and suitable approach to measuring neuronal calcium signals, the authors should at least discuss the possibility that measuring calcium photometry signals at the cell soma region might not reflect neuronal firing activity (Legaria et al., 2022), and thus this activity might not indicate rILN neuron signaling to the striatum. Fiber photometry recordings of the axons of these projection neurons in DS would provide more definitive evidence for this signaling.

Therefore, it would be helpful if the authors could tune down the conclusions related to these areas.

Other suggestions include:

– Please clarify if DS projecting neurons from all rILN nuclei show the reported action initiation and reward acquisition activity or only CL neurons.

– Along similar lines, to what extent of rILN was targeted for optogenetic activation and inhibition? It seems that the authors implanted a total of 4 optic fibers, two on each side (please clarify in methods). What was the reasoning behind this? Please show that only rILN and not PF was activated/inhibited.

– The transsynaptic tracing experiments – the cell count quantifications in CM, PC, and CL are needed.

– Why is the injection site for the retrograde cre-dependent tdTomato AAV (Figure 5 middle left panels) showing expression? Please clarify. Is the cre coming through transsynaptic AAV1 from direct projections of each AAV1 injection site (AAV1 is not supposed to spread across a second synapse)? The diagrams suggest that not all regions (e.g. SUM or SC) have direct projections to DS.

*Reviewer #2 (Recommendations for the authors):*

I think it was very well conducted and is very strong, well written, and within the scope of eLife's typical papers. I look forward to seeing where the authors take this line of investigation.

*Reviewer #3 (Recommendations for the authors):*

1) The final neuroanatomical tracing studies are not very well integrated with the rest of the study. The experiments are introduced as a comparison between the striatum- and cortex-projecting neurons, but the latter experiment was not done. Without distinguishing these two populations or possibly comparing the inputs to intralaminar versus parafascicular nuclei, the experiment does not offer a lot to the overall conclusions. The authors should either do the additional work to make a meaningful conclusion from these experiments or exclude them from the current manuscript.

2) In line 290, the authors state that "in addition to dopamine terminal signaling coinciding with reward presentation (Howe & Dombeck, 2016); an activity that is not reflected in cell body firing", however, there are plenty of examples of reward-responsive VTA dopamine neurons from single-unit recordings in SNc/VTA (e.g., PMID: 22258508). The authors should also be very careful in their interpretation of the cited work. The Howe and Dombek paper do not claim that movement and reward-related signals are not encoded in the midbrain, but rather that they are differentially encoded by distinct dopamine neurons subpopulations (see their more recent work: bioRxiv: 2022.06.20.496872) arguing that the differences between axonal and somatic dopamine cell responses is due to genetic identity rather than terminal modulation. Undoubtedly terminal modulation is a key player, and this idea is recently gaining a lot of traction, but the dopamine signal in the striatum likely reflects a combination of somatic activity and terminal modulation, and not just the latter.

3) The authors show that activity in rILN→striatum projection neurons correlates with movement in S2, and their previous study showed that chemogenetically inhibiting all rILN neurons suppresses movement. How do the optogenetic manipulations here affect spontaneous movement and might this explain their observed effects on goal-directed operant responding in Figure 4?

[Editors' note: further revisions were suggested prior to acceptance, as described below.]

Thank you for resubmitting your work entitled "The rostral intralaminar nuclear complex of the thalamus supports striatally-mediated action reinforcement" for further consideration by *eLife*. Your revised article has been evaluated by Kate Wassum (Senior Editor) and a Reviewing Editor.

The manuscript has been improved but there are some remaining issues that need to be addressed, as outlined below:

Reviewer #3 has raised some concerns about Figure 5. Please revise the figure and related text to make the figure/manuscript more clear.

*Reviewer #1 (Recommendations for the authors):*

This revised manuscript has addressed all the concerns I raised in the previous round of review.

*Reviewer #2 (Recommendations for the authors):*

In my previous review, I did not have major issues with the manuscript at large. Looking over the other reviewers' comments and the response to the reviewers, it largely seems that the authors have addressed concerns with clarification and some inclusion of new data. I still find this work to be interesting and compelling as I did previously.

*Reviewer #3 (Recommendations for the authors):*

In Figure 5C (left), it isn't clear that the fluorogold/AAV1 injection was actually into the supramamillary nucleus. It's hard to tell where the injection was but it seems to be dispersed throughout the midbrain.

Also, in Figures 5A, B, and D, the way the authors have schematized ipsi/contralateral projections is confusing, in particular, showing three circles with the seed nucleus. Perhaps the authors can think of a better way to convey laterality.

---

## [Author Response]

Essential revisions:Reviewer #1 (Recommendations for the authors):While overall this is a well-executed study that uses a variety of behavior tasks to demonstrate the activity of DS-projecting rILN neurons. Yet, the different parts of this study (electrophysiology, behavior, tracing) feel somewhat disjointed, and additional control experiments would be needed to substantiate some of the claims made.– The statement that the authors "discovered that rILN neurons projecting to the DS integrated information from several subcortical and cortical sources" (abstract lines 39 – 41) is misleading and not fully supported by the presented data. While the authors demonstrate that rILN neurons can receive input projections from several subcortical and cortical areas, they do not show whether these projections converge onto the same individual neurons in rILN, and more importantly, they do not show whether "information" is carried by these projections or integrated into rILN neurons during behavior. Recordings and manipulation experiments of these subcortical and cortical inputs to rILN would be necessary to support such a claim.

Thank you for identifying this inadvertent misrepresentation in our conclusions. We have revised this sentence to state: “Here we discovered that rILN neurons projecting to the DS are innervated by a range of cortical and subcortical afferents and that rILNDS neurons stably signaled at two time points in mice performing an action sequence task reinforced by sucrose reward: action initiation and reward acquisition” (lines 39 – 42).

– While fiber photometry is a powerful and suitable approach to measuring neuronal calcium signals, the authors should at least discuss the possibility that measuring calcium photometry signals at the cell soma region might not reflect neuronal firing activity (Legaria et al., 2022), and thus this activity might not indicate rILN neuron signaling to the striatum. Fiber photometry recordings of the axons of these projection neurons in DS would provide more definitive evidence for this signaling.Therefore, it would be helpful if the authors could tune down the conclusions related to these areas.

Thank you for bringing our attention to this publication, which is now cited in our discussion. We now include acknowledgement of this potential limitation with photometric recordings in our manuscript (lines 292 – 296).

Other suggestions include:– Please clarify if DS projecting neurons from all rILN nuclei show the reported action initiation and reward acquisition activity or only CL neurons.

The arrangement of the rILN nuclei presents a technical challenge for experiments attempting to selectively record from or manipulate a single nucleus in this grouping. Based on our findings that the three nuclei do not differ in electrophysiological properties, we approached the in vivo experiments with the intent to target the rILN as a unit. As the reviewer points out, the medial-lateral coordinate for optic fiber placement tended to align above the CL and PC nuclei. However, variability in fiber placement and spread of light within tissue resulted in inclusion of CM activity as well. Given the spread of light through tissue (Shin, et al., 2016; PMID: 27895987), it would be very difficult to confidently determine from histology which photometry recordings were primarily obtained from CL vs PC vs CM neuronal activity. We agree with the reviewer that these three nuclei may differently signal during reward-driven behavior. Our di-synaptic tracing study supports this possibility as it revealed unique afferent connectivity to rILNDS projecting neurons. We now mention this limitation of our approach in the discussion (lines 324 – 330).

– Along similar lines, to what extent of rILN was targeted for optogenetic activation and inhibition? It seems that the authors implanted a total of 4 optic fibers, two on each side (please clarify in methods). What was the reasoning behind this? Please show that only rILN and not PF was activated/inhibited.

We apologize for the confusion in our description of this method. For our optogenetic experiments, we infused viruses at four locations (bilateral striatum and rILN) and implanted only two fibers (bilateral rILN) to selectively target striatally-projecting rILN neurons. We have added clarification on this detail to the methods section.

To prevent inadvertent modulation of Pf neurons, we used virus injection coordinates and volumes that prevented viral spread to the Pf and furthermore implanted the optic fibers in the more rostral regions of the rILN. We histologically confirmed viral expression and fiber placement for all mice and excluded any mice with viral spread to the Pf or off-target fiber placement. We include these criteria for post-hoc exclusion in the methods.

– The transsynaptic tracing experiments – the cell count quantifications in CM, PC, and CL are needed.

Thank you for this suggestion, we now include cell counts for 2 cases per investigated afferent (Figure 5 Supplement 2).

– Why is the injection site for the retrograde cre-dependent tdTomato AAV (Figure 5 middle left panels) showing expression? Please clarify. Is the cre coming through transsynaptic AAV1 from direct projections of each AAV1 injection site (AAV1 is not supposed to spread across a second synapse)? The diagrams suggest that not all regions (e.g. SUM or SC) have direct projections to DS.

We apologize for this confusion. The tdTomato fluorophore expression observed in the striatum may arise from several possible circuit configurations. To survey just a couple: (1) tdTomato expression in the DS arises from direct projections from the afferent bypassing the thalamus (e.g. ipsilateral ACCStriatum), which would result in labeled striatal somata (ACC pyramidal neurons delivering AAV1-cre to an MSN, and those local MSN collaterals retrogradely picking up rAAV-DIO-tdtomato) and ACC labeled axon terminals in the DS (ACC interneurons delivering AAV1-cre to DS-projecting ACC pyramidal neurons that pick up rAAV-DIO-tdtomato); (2) terminal projections arising from the labeled rILN neurons shown in the middle-right panels (i.e. ACCrILNStriatum).

Reviewer #3 (Recommendations for the authors):1) The final neuroanatomical tracing studies are not very well integrated with the rest of the study. The experiments are introduced as a comparison between the striatum- and cortex-projecting neurons, but the latter experiment was not done. Without distinguishing these two populations or possibly comparing the inputs to intralaminar versus parafascicular nuclei, the experiment does not offer a lot to the overall conclusions. The authors should either do the additional work to make a meaningful conclusion from these experiments or exclude them from the current manuscript.

Thank you for identifying this weakness in our introduction for the tracing study. We agree that it was inadvertently misleading to reference the rILNcortical projection without including comparable experimental results. As such, we have rewritten our rationale for the experiment in the manuscript (lines 220-222).

We do feel that this experiment assessing rILNstriatal afferents has merit and sets the foundation to assess differences between rILN subpopulations with unique postsynaptic targets. Given that our manuscript exclusively characterizes the rILNstriatal projection in both physiology and behavior, we thought it was fitting to include a qualitative survey of afferents that may contribute to the behaviorally dynamic activity of this circuit. Whereas future investigators may seek to compare the connectivity and function of the rILN with other thalamic nuclei, such as the parafascicular nucleus, we respectively feel that it extends beyond the scope of the current study.

2) In line 290, the authors state that "in addition to dopamine terminal signaling coinciding with reward presentation (Howe & Dombeck, 2016); an activity that is not reflected in cell body firing", however, there are plenty of examples of reward-responsive VTA dopamine neurons from single-unit recordings in SNc/VTA (e.g., PMID: 22258508). The authors should also be very careful in their interpretation of the cited work. The Howe and Dombek paper do not claim that movement and reward-related signals are not encoded in the midbrain, but rather that they are differentially encoded by distinct dopamine neurons subpopulations (see their more recent work: bioRxiv: 2022.06.20.496872) arguing that the differences between axonal and somatic dopamine cell responses is due to genetic identity rather than terminal modulation. Undoubtedly terminal modulation is a key player, and this idea is recently gaining a lot of traction, but the dopamine signal in the striatum likely reflects a combination of somatic activity and terminal modulation, and not just the latter.

Thank you for pointing out our over-interpretation of these studies, we have modified our discussion to emphasize that both dopamine neuron somatic activity and terminal modulation likely contribute to reward-related signaling (lines 283 – 289).

3) The authors show that activity in rILN→striatum projection neurons correlates with movement in S2, and their previous study showed that chemogenetically inhibiting all rILN neurons suppresses movement. How do the optogenetic manipulations here affect spontaneous movement and might this explain their observed effects on goal-directed operant responding in Figure 4?

This a great question. We analyzed a number of parameters in the FR5 task to examine whether optogenetic inhibition altered movement. 2-way repeated measures ANOVAs (factors of light delivery and virus expression) did not reveal significant effects for the average FR5 inter-press interval (light: F(1,26)=1.45, P=0.24; virus: F(1,26)=0.33, P=0.57), total FR5 duration (light: F(1,26)=0.001, P=0.97; virus: F(1,26)=0.22, P=0.65), or reward retrieval latency (light: F(1,26)=0.80, P=0.34; virus: F(1,26)=0.99, P=0.33). We did find a significant effect of light delivery on the latency to initiate the FR5 sequence (an average increase of 100 ms for NpHR-eYFP mice and 160 ms for eYFP control mice; light: F(1,26)=17.37, P=0.0003; virus: F(1,26)=0.05, P=0.83), but only the eYFP control cohort yielded a significant post-hoc test (P = 0.0033; NpHR-eYFP: P = 0.054).

We think two factors contribute to this apparent discrepancy in motor impairment between the present experiment and the published Cover et al., 2019 Cell Reports finding. The 2019 experiment chemogenetically suppressed somatic and terminal activity of all rILN neurons, whereas here we transiently inhibited somatic activity of striatally projecting rILN neurons. Beyond this methodological difference, however, we suspect that the context underlying movement may play a significant role. The 2019 study monitored spontaneous locomotion in an environment free of rewards and meaningful sensory information. In contrast, the present study assessed performance on an external cue-driven paradigm in motivated (food-restricted) subjects.

[Editors' note: further revisions were suggested prior to acceptance, as described below.]

Reviewer #3 (Recommendations for the authors):In Figure 5C (left), it isn't clear that the fluorogold/AAV1 injection was actually into the supramamillary nucleus. It's hard to tell where the injection was but it seems to be dispersed throughout the midbrain.

The core of the injection site does appear in the area of the SUM, as evidenced by the dense fluorogold labeling. The purplish, scattered off-target labeling appears associated with histological imperfections resulting from our mounting method. The prominent red (tdtomato) labeling lateral to the injection site (midbrain/substantia nigra) are tdtomato-positive axons originating from the dorsal striatum. After further assessment, we now cannot be certain that the injection site did not also involve the lateral hypothalamus, however, which sits laterally to the SUM. As such, we have revised the manuscript and associated figure/figure legends to reflect this interpretation. Where we used SUM to abbreviate for supramammillary nucleus, we now used S/L to abbreviate for supramammillary nucleus/lateral hypothalamus. Thank you for pointing this out

Also, in Figures 5A, B, and D, the way the authors have schematized ipsi/contralateral projections is confusing, in particular, showing three circles with the seed nucleus. Perhaps the authors can think of a better way to convey laterality.

We have revised the way ipsi/contralateral projections are schematized. We now just show two circles: one depicting the ipsilateral seed nucleus and the other the contralateral seed nucleus. We revised the figure legend accordingly. We agree that this streamlines conceptual understanding for the reader. Thank you for the suggestion.